# Real-Time Object Detection from UAV Inspection Videos by Combining YOLOv5s and DeepStream

**DOI:** 10.3390/s24123862

**Published:** 2024-06-14

**Authors:** Shidun Xie, Guanghong Deng, Baihao Lin, Wenlong Jing, Yong Li, Xiaodan Zhao

**Affiliations:** 1Guangdong Engineering Technology Research Center of UAV Remote Sensing Network, Guangzhou iMapCloud Intelligent Technology Co., Ltd., Guangzhou 510095, China; xieshidun@geoai.com (S.X.); linbaihao@geoai.com (B.L.); zhaoxiaodan@geoai.com (X.Z.); 2Guangdong Province Engineering Laboratory for Geographic Spatiotemporal Big Data, Key Laboratory of Guangdong for Utilization of Remote Sensing and Geographical Information System, Guangdong Open Laboratory of Geospatial Information Technology and Application, Guangzhou Institute of Geography, Guangdong Academy of Sciences, Guangzhou 510070, China

**Keywords:** UAVs, YOLOv5, object detection, DeepStream, route planning

## Abstract

The high-altitude real-time inspection of unmanned aerial vehicles (UAVs) has always been a very challenging task. Because high-altitude inspections are susceptible to interference from different weather conditions, interference from communication signals and a larger field of view result in a smaller object area to be identified. We adopted a method that combines a UAV system scheduling platform with artificial intelligence object detection to implement the UAV automatic inspection technology. We trained the YOLOv5s model on five different categories of vehicle data sets, in which mAP50 and mAP50-95 reached 93.2% and 71.7%, respectively. The YOLOv5s model size is only 13.76 MB, and the detection speed of a single inspection photo reaches 11.26 ms. It is a relatively lightweight model and is suitable for deployment on edge devices for real-time detection. In the original DeepStream framework, we set up the http communication protocol to start quickly to enable different users to call and use it at the same time. In addition, asynchronous sending of alarm frame interception function was added and the auxiliary services were set up to quickly resume video streaming after interruption. We deployed the trained YOLOv5s model on the improved DeepStream framework to implement automatic UAV inspection.

## 1. Introduction

Real-time object detection is not only a challenging task in computer vision, but also a hot topic in industrial applications, for example, in object tracking [1], autonomous driving [2], medical image processing [3], agricultural machine vision applications [4], etc. Real-time detection requires lightweight convolutional neural networks and equipment that can process floating point operations faster. Equipment that implements real-time detection usually includes mobile GPU or CPU servers and various neural processing units (NPUs). Different edge processing machines focus on the acceleration of different modules. In this paper, we propose a real-time detection system mainly applied to cloud or mobile devices.

In recent years, different real-time detection models have been proposed, suitable for different edge devices. MCUNet [5] and NanoDet [6] are mainly designed to produce lower-power microcontrollers and improve edge CPU inference speed. The You Only Look Once (YOLO) series of algorithms have better accuracy and faster reasoning speed, and are widely used in the industrial field. YOLOv1 [7] is a typical one-stage object detector, and based on this, a series of improvements were made, resulting in YOLOv2 [8], and YOLOv3 [9], which have faster detection speeds and higher detection accuracy. YOLOv4 [10] redesigns the three independent architectures of the trunk, neck and head to make them better trained on a single GPU. At present, YOLOv5 [11], YOLOX [12], PPYOLOE [13], etc., have extremely competitive performance in real-time detection and deployment. More recently, real-time object detectors have mainly focused on simple and efficient structural design, so that real-time detection effects can be achieved when used on the CPU [14,15,16]. The simple framework design with good performance is mainly based on MobileNet [17,18,19], ShuffleNet [20,21], or GhostNet [22]. However, real-time detectors developed based on GPU [23,24] mostly use ResNet [25] or DLA [26], combined with the CSPNet [27] strategy to optimize the architecture and improve detector performance.

With the continuous development of lightweight real-time detection models, some types of industrial application equipment accompanied by algorithms are also constantly being introduced. The development and application of UAVs have made many fields more intelligent, such as industrial inspection [28], intelligent substation inspection [29], agricultural applications [30], etc. Running convolutional neural networks on embedded systems has become a reality, and when combined with the application innovation of UAVs, there will be new ways of application in various fields. Tijtgat et al. [31] designed a system based on NVIDIA Jetson TX2 edge computing device running YOLOv2 to achieve real-time object detection with UAVs. Abdulghafoor et al. [32] proposed a method of combining edge computing devices with a DeepStream software development kit (DS-SDK) 4.0.2 [33] to implement a convolutional network model that can process video streams with high performance. In order to improve the practicality of a real-time video stream detection system, Guo et al. [34] proposed a novel region of interest detection (ROIDet) algorithm and designed a bandwidth-efficient multi-camera video streaming system for deep learning video analysis. Hossain et al. [35] proposed the joint implementation of the application of deep learning technology with a computer system integrated with the UAV, which can track and detect objects in real time. Vandersteen et al. [36] proposed a multi-data set learning strategy to optimize the real-time performance of detection on embedded hardware devices and improve detection efficiency. Haq et al. [37] deployed the DeepStream framework on the NVIDIA Jetson single-board computer to run deep learning algorithms, especially the YOLO algorithm. The study also verified that the DeepStream framework can run well in virtual machines, especially using Docker, which can further improve the performance of the model and the portability during the deployment process. Huu et al. [38] proposed a method based on the NVIDIA DS-SDK architecture, using multiple surveillance camera detection methods to implement the application of deep learning-based algorithms for vehicle monitoring. Ghaziamin et al. [39] deployed the object detection model to Nvidia Jetson devices and designed a passenger counting system. And after edge deployment through Nvidia DeepStream, it improved efficiency while saving the use of hardware resources. Smink et al. [40] used edge devices combined with the detection and tracking system of the NVIDIA DeepStream framework to implement a set of real-time tag-reading applications. Qaraqe et al. [41] designed an end-to-end security intelligent monitoring system that used the DeepStream software development kit (SDK) for real-time inference, which can have a significant impact on public safety and crowd management.

In summary, their method cannot simultaneously ensure detection accuracy and must also maintain detection speed. Especially in the UAV inspection process, the requirements for hardware robustness are relatively high. Existing real-time inspection technology, especially the use of fixed camera equipment for monitoring, will result in a small field of view, poor mobility of the equipment, the inability to realize real-time switching between different areas, and the existence of large monitoring blind spots and other problems. UAV high-altitude inspection can only solve the problems of vision and flexibility. Therefore, this article proposes a method that combines UAVs and artificial intelligence real-time detection to implement an automatic UAV real-time inspection system. The main contributions of this article are as follows:The integration of UAV inspection, YOLOv5s object detection, and DeepStream framework realizes a new real-time object detection method.The DeepStream service can be quickly started using the http communication protocol, and can be called and used by different users at the same time.The asynchronous sending of the alarm frame interception function is implemented, making the real-time video stream smoother.Auxiliary services can be set up to quickly resume video streaming after interruption.

## 2. Methods

### 2.1. DeepStream Software Development Kit

DS-SDK [42] is an intelligent video analysis suite assembled based on NVIDIA technology. It introduces deep neural networks and other complex processing tasks into stream processing pipelines to achieve near real-time analysis of video and other sensor data. The application framework has hardware-accelerated processing building blocks, so developers do not need to design an end-to-end solution from scratch and need only to focus on building the core deep learning network and video stream processing modules. Moreover, it builds an end-to-end video stream detection pipeline based on gstream, which simplifies the development and application of video streams. Algorithm personnel can quickly convert other video stream processing capabilities while focusing on the development of all components. The original DeepStream flow chart is shown in Figure 1.

In Figure 1, the basic workflow of DeepStream’s original architecture is shown. First, we need to start a DeepStream video streaming service on the backend of the platform, immediately start creating the DeepStream pipeline, and determine the pipeline initialization status. If the pipeline initialization verification is successful, then the pipeline status is set to the playing state, which is the pipeline starting working status. Secondly, the pipeline starts to push the stream, and the algorithm performs video stream data pre-processing including normalization and scaling. Then, the video stream will be input into the YOLOv5 model for inference, and finally the DeepSORT [43,44] tracking algorithm is used to track the corresponding object. Add alarm logic to the object that needs to be identified, push the processed video stream to the platform service, and finally display the identification results in real time on the platform software 2.5.9.

The original DeepStream framework is connected to a UAV inspection video, and users cannot call different services and different recognition algorithms at any time during use. The communication protocol does not support UAV video streaming communication services. Therefore, we improved DeepStream, as shown in Figure 2.

As shown in Figure 2, this shows the improved workflow of the DeepStream architecture. First, the platform starts the DeepStream service command by sending an http request. Currently, once the algorithm service receives the platform request to start the command, it immediately starts the DeepStream service, starts parsing the request command parameters, determines whether the parameters meet the requirements, and generates a configuration file according to the corresponding task. Secondly, after the configuration is successful, start creating a sub-process to perform the task and create a DeepStream pipeline. Determine whether the pipeline is initialized. After the initialization is successful, set the pipeline status to the playing state. The pipeline starts pushing and the algorithm starts working, including pre-processing, reasoning, tracking, and other operations. After the object is detected, alarm logic is added. Finally, the video stream is pushed to the platform service, and the platform interface displays the stream inferred by the algorithm. In addition, we also added an auxiliary service, whose purpose is to monitor whether the DeepStream service is disconnected. If a disconnection occurs, the auxiliary service will immediately kill the task and restart the task immediately.

### 2.2. The YOLO Algorithm

The YOLOv5 algorithm is an outstanding object detection algorithm and has good robustness to UAV inspection images. YOLOv5 contains five network models of different sizes, namely YOLOv5n, YOLOv5s, YOLOv5m, YOLOv5l and YOLOv5x. This article verifies the performance of different YOLOv5 models and selects the most suitable detector for deploying the DeepStream architecture. The algorithm structure flow chart is shown in Figure 3.

As shown in Figure 3, there are two extremely important elements, data processing and data training. The first is the data processing stage. In the process of collecting data, we need to plan the UAV inspection route. In the areas that need to be inspected, let the UAVs automatically inspect and collect photos or videos according to the planned route. During the process of collecting images with a UAV, we flew in different time periods, different weather, and different lighting conditions to achieve a better performance of the model. To make the model more robust to the complex background of high-altitude inspections, we planned the flight area to include different areas such as villages, water bodies, and towns. The collected UAV inspection videos need to be manually edited and the required object information retained and extracted into images. Images were manually filtered and labeled using LabelImg software 1.8.6. Finally, the data were divided into a training set and a test set.

The second stage is that of model training. In the first stage of data processing, the processed data were divided into a training set and test set. We were in the data collection phase. Effectively planning routes for different scenarios and collecting data under different conditions fully ensures the richness of model training samples and can effectively prevent over-fitting during the training process. We evaluated the performance of YOLOv5n, YOLOv5s, YOLOv5m, YOLOv5l, and YOLOv5x models in the process of UAV high-altitude inspection. The performance was evaluated on the training set and test set divided from the same data set. Finally, we selected a model with strong robust performance for real-time detection based on its real-time detection effect.

Since UAVs fly at a high altitude, they will keep shaking, the field of view when flying at high altitude is large, and the shooting area is also large, resulting in a smaller target to be detected. There are difficulties in using object detection algorithms to detect smaller objects. This article mainly used YOLOv5s, a lightweight object detection model, combined with DeepStream architecture to achieve real-time video stream detection. To better implement a complete set of real-time target detection system architecture, we made a series of improvements to the original DreamStream and combined with the YOLOv5s model to achieve target detection. YOLOv5s is designed to detect the goals that require inspection. It is designed to achieve automatic detection capabilities. The main improvement part is the design of the overall logical architecture of DeepStream, and the improved experimental results are also very effective.

### 2.3. Evaluation Indicators

Due to the high-altitude inspection of UAVs, as the height of the UAV flight increases, the inspection field of view becomes wider. It is easily affected by the complex low-altitude environment, causing certain false and missed detections. Therefore, we used precision and recall measuring to assess whether a detector had good performance. The precision and recall formulas are shown in (1) and (2):(1)$$P=\frac{TP}{TP+FP}$$
(2)$$R=\frac{TP}{TP+FN}$$
where P and R represent precision and recall, respectively. TP represents the true positives; FP represents the false positives; and FN represents the false negatives. 

The F_1_ value combines the two indicators of precision and recall to comprehensively evaluate the effect of the model. As shown in Formula (3).
(3)$$F_{1}=\frac{2PR}{P+R}$$

In order to further obtain a better real-time video stream detection model, two evaluation indicators, *AP* (average precision) and *mAP* (mean average precision), were used. The Formulas (4) and (5) of *AP* and *mAP* are shown as follows:(4)$$AΡ=\int_{0}^{1}Ρ\left(R\right)dR$$
(5)$$mAΡ=\frac{\sum_{q=1}^{Q}AΡ\left(q\right)}{Q}$$
where Q represents the number of categories.

## 3. Experiment and Results

### 3.1. Experimental Conditions

#### 3.1.1. Data Acquisition and Transmission

This article mainly collected the data we needed through UAV inspection. A UAV can achieve automatic flight collection through the automatic command and dispatch platform. The command and dispatch interface are shown in Figure 4.

It can be seen from Figure 4 that the flight of a UAV realizes automatic inspection through the dispatch of the command and dispatch system. Through the command and dispatch system interface, the status of the UAV and its slot can be monitored in real time. We can see a white drone nest, the drone nest door has been opened, the platform has been raised, and a black UAV is waiting to fly on it. There are three buttons available on the left side of the command and dispatch system. The main functions include UAV control, landing control, system functions, power control, emergency control, etc. It can not only command the takeoff and landing of the drone, but also control the angle and focus of the gimbal. When the drone is flying, the battery power can be monitored in real time. When an emergency occurs, such as low battery or automatic hovering, it can also return to home with one click.

The data used in this article mainly use images or videos collected during UAV inspections to annotate them. The annotated original image is shown in Figure 5. This article used LabelImg to label five different categories of vehicle models. The labels are sjc, hc, wjj, dzj, and dc. Where sjc represents a car, hc represents a truck, wjj represents an excavator, dzj represents a pile driver, and dc represents a crane. To ensure the effectiveness of model training in this article, we increased the richness of samples as much as possible. Therefore, in the process of collecting images, we planned routes in different areas and different route altitudes. While ensuring flight safety, the drone’s flight speed and different flight mileage were set.

Figure 6 shows the data set used for training in this article, with a total of 3208 images. There were 2887 images in the training set and 321 images in the test set. We divided the training set and the test set at a ratio of 9:1. To ensure the effectiveness of the training model, the divided data sets were independent of each other.

#### 3.1.2. Inspection Routes

To realize the automatic take-off and landing flight of a UAV, you first need to plan an inspection route. We planned four different UAV inspection routes as shown in Figure 7, and the parameters of UAV inspections on different routes are shown in Table 1.

It can be concluded from Figure 7 that in order to realize automatic UAV inspection, we planned four inspection routes on the UAV command and dispatch platform. Among them, the blue linear area is the area where the UAV needs to fly according to the route.

The routes are all included in a yellow circular area, which is displayed on a map interface. We can clearly see that there are residential areas and river areas next to the planned routes. To increase the richness of the sample, our routes span multiple regions to collect images. Table 1 shows the flight parameters of four different inspection routes, such as flight length, flight duration, route altitude, round-trip speed, route speed, take-off and landing altitude, etc.

#### 3.1.3. Experiment Platform

Server-side: Ubuntu 18.04, Intel^®^ Silver 4210 CPU@2.20 GHz, NVIDIA GeForce RTX A100(80 GB) GPU. The model framework is Pytorch 1.10.0, and the related software is CUDA11.1, CUDNN 8.0.5 and Python 3.8.

Ubuntu 20.04, Intel^®^ Xeon^®^ Gold 6278C CPU@2.60 GHZ, NVIDIA Tesla T4(16G) GPU. The related software is CUDA 11.6, CUDNN 8.4.0.

### 3.2. Experimental Results

#### YOLOv5 Detection Results

After the model training was completed, to verify the detection effects of different models, we used images containing five different categories to test the detection performance of five different models. The detection results of different models are displayed in Figure 8.

From Figure 8, we can obtain the detection results of five different YOLOv5 models. By comparing the results, it can be concluded that YOLOv5n can, to a certain extent, identify five different categories of vehicles. However, the roof of the building was misidentified. YOLOv5s, YOLOv5m, YOLOv5l, and YOLOv5x can all produce better detection results. We can clearly see that different YOLOv5 detection models can detect five different types of objects, namely, car, truck, excavator, pile driver, and crane. This experiment shows the effectiveness of the training method in this paper and its ability to accurately detect different types of vehicles.

### 3.3. Comparative Requirements

In order to further verify the robust performance of this model, we compared two different models, mAP50 and mAP50-95, as shown in Table 2 and Table 3, respectively.

It can be seen from Table 2 that the mAP50 of YOLO5n, YOLOv5s, YOLOv5m, YOLOv5l, and YOLOv5x reached 90.1%, 93.2%, 93.7%, 94.2%, and 94.7%, respectively. Among them, the mAP50 detection accuracy of the YOLOv5s model is 3.1% higher than that of YOLOv5n, but the difference in mAP50 accuracy with YOLOv5m, YOLOv5l, and YOLOv5x is minimal.

It can be seen from Table 3 that the mAP50-95 of YOLO5n, YOLOv5s, YOLOv5m, YOLOv5l, and YOLOv5x reached 64.5%, 71.7%, 74.8%, 76.6%, and 77.6%, respectively. Among them, the detection accuracy of mAP50-95 of the YOLOv5n model is extremely low. This experiment shows that YOLOv5n is less robust than the other four trained models.

To train a better real-time object detector, we must not only compare its detection accuracy but also its detection speed. Therefore, the detection speeds of different models are shown in Table 4.

Table 4 shows the detection speed of YOLOv5n, YOLOv5s, YOLOv5m, YOLOv5l, and YOLOv5x on NVIDIA A100 GPU, which are 11.1 ms, 11.26 ms, 15.77 ms, 17.44 ms, and 20.29 ms, respectively. The model sizes of YOLOv5n, YOLOv5s, YOLOv5m, YOLOv5l, and YOLOv5x are 3.7 MB, 13.76 MB, 40.27 MB, 88.57 MB, and 165.13 MB, respectively. Compared with YOLOv5n, the inference speed of YOLOv5s is 0.16ms slower, but the inference speed is faster than YOLOv5s, YOLOv5m, YOLOv5l, and YOLOv5x. To verify the performance of a model during the training process, you can observe whether the model’s loss gradually converges. The training loss and the verification loss of this model are shown in Figure 9.

It can be concluded from Figure 7 that during the training period of 500 epochs, the training loss and verification loss of YOLOv5n, YOLOv5s, YOLOv5m, YOLOv5l, and YOLOv5x gradually decreased and reached a certain degree of convergence. At the same time, the effectiveness of the training method in this article is verified.

The accuracy of a model in detecting objects is measured during the training process, and the mAP50 and mAP50-95 of the model can be tested during the iteration process. The performance of mAP50 and mAP50-95 of the YOLOv5 model during the training iteration process is shown in Figure 10.

It can be concluded from Figure 10 that the mAP50 detection accuracy of the YOLOv5s model trained in this article is significantly higher than the YOLOv5n model in 500 epochs of training but is not much different from the mAP50 performance of the YOLOv5m, YOLOv5l, and YOLOv5x models. However, the mAP50-95 performance of the YOLOv5s model is significantly better than the YOLOv5n model, but lower than the YOLOv5m, YOLOv5l, and YOLOv5x models.

The confusion matrix is a common visualization tool for measuring the performance of object detectors. You can clearly see the performance of the various indicators in Figure 11. It shows the confusion matrix diagrams of the five different models, YOLOv5n, YOLOv5s, YOLOv5m, YOLOv5l, and YOLOv5x, respectively.

It can be seen from Figure 11 that the data in each grid of the confusion matrix are the result of normalization. We can measure the performance capability of a model only by comparing the size of the diagonal value. The five category values of YOLOv5s are 0.92, 0.80, 0.97, 0.99, and 1.00, respectively. The results of the YOLOv5n model tested in five different categories were 0.87, 0.71, 0.96, 0.98, and 0.99, respectively. Therefore, the YOLOv5s model performance is significantly better than YOLOv5n. At the same time, it can also be concluded that compared with YOLOv5m, YOLO v5l, and YOLOv5x, the performance gap is not obvious.

The precision and recall curves are a basic indicator of the robustness of a model. During the process of training and testing the model, when the precision is close to zero, the recall is close to 1. When the recall is close to 0, the precision is close to 1, as shown in Figure 12.

Figure 12 shows the precision and recall curves of five different models. It can be clearly seen from Figure 12 that the fitting curves of five different models of YOLOv5 all show good results. This experiment shows the effectiveness of the training method in this article.

In order to combine precision and recall performance metrics, this paper also shows the F_1_ value to measure the performance of the detector, and the confidence level to measure the probability of a object prediction. The F_1_ value and the confidence curve are shown in Figure 13.

It can be seen from Figure 13 that the F_1_ values of five different models change as the confidence level changes. It shows the F_1_ score performance under different confidence classification thresholds. It can be seen from the figure that the model in this article also has a higher F_1_ value at a higher confidence level, indicating that the model trained in this article has higher robustness.

We trained and tested five different YOLOv5 models on the same data set, and finally selected the YOLOv5s model, which is relatively lightweight. to the correct selection of a data set is extremely important for model training. Therefore, in order to further verify the effectiveness of the design of this article, Figure 14 shows the correlation of labels.

Figure 14 shows the correlation distribution diagram of the data set labels we used. From the figure, we can clearly see the distribution of the center point, width, and height of the image annotation target. It shows that the labeled data samples used by this algorithm are diverse and the samples are very rich. It is effective for model training.

### 3.4. UAV Inspection Interface Display

During the flight inspection process of the UAV, the object detection algorithm service is called to deploy the trained model on an NVIDIA Tesla T4 GPU cloud server. We deployed a set of vehicle detection methods on this server using a set of algorithms called by the DeepStream framework. When the UAV conducts inspections along the designated route, we can clearly see the detection results processed by the UAV in real time on the dispatch platform. Figure 15 shows the real-time detection results of the UAV during automatic inspection according to the planned route. 

In Figure 15, a display of real-time inspection pictures is shown. During this type of high-altitude inspection by drones, the drones are looking down at the ground angle. Information such as buildings, ponds, and vehicles on the ground can be clearly seen. It can also be clearly seen that the YOLOv5s model trained in this article can automatically detect the required targets during the automatic drone inspection process, and correctly distinguish the target area and the background area. In the lower right corner of the image, you can obtain the inspection route of the drone and the specific location information of the drone flying on this route.

### 3.5. Mathematical Analysis of DeepStream Service Resources and Startup Time

To verify the quality of a framework, we not only consider its algorithm detection capabilities, but also consider its reasonable utilization of effective resources. Therefore, it is necessary to compare the real-time video streams before and after the improvement to check the resource overhead of the system architecture. The main comparison before and after improvement is as follows:Before the improvement, it was necessary to bind drone equipment and specify execution tasks. In terms of resource consumption, a Telsa T4 GPU, 16G graphics card server can only bind up to six devices and tasks to be executed. That is, the binding of tasks limits the reasonable utilization of GPU and memory resources and cannot be dynamically adjusted according to task requirements. In short, the performance of the hardware before the improvement was very poor.After improvement, our real-time video stream detection system does not need to bind devices and tasks; tasks are no longer bound, and resource allocation can be dynamically adjusted according to real-time needs. Not only can multiple devices be controlled at the same time, but different tasks can be switched freely. Therefore, after the improvement, the flexibility and the scalability of the system have been greatly improved.

DeepStream startup time mainly consists of two time elements. One is the time from the request to the pipeline initialization, and the other is the time from the pipeline initialization to the pipeline state switching to the playing state. The sum of the two is the startup time. The initial application of the DeepStream service has a startup time of approximately 7 s. Therefore, there is huge room for improvement. Based on this, we tested the startup of video streams in different definitions. The definition is divided into five levels, namely ultra-high definition (Ultra HD), ultra definition (UD), high definition (HD), standard definition (SD), and smooth, as shown in Table 5.

As shown in Table 5, during the first test, the first-stage startup time of UHD was 2.21 s, and the second-stage startup time was 0.60 s. The total time consumed is 2.81 s. The total time for the second test of UHD video is 2.52 s. The overall consumption time is significantly faster than other low-definition videos. Moreover, the startup time of the optimized model architecture is significantly faster than the original DeepStream processing time.

## 4. Conclusions

In this paper, we trained the YOLOv5s model to detect five different types of vehicles, and combined UAV automatic inspection technology to achieve the automatic real-time detection of UAVs. The main contributions of this article are as follows:The integration of UAV inspection, YOLOv5s object detection, and the DeepStream framework realizes a new real-time object detection method.The DeepStream service can be quickly started using the http communication protocol, and can be called and used by different users at the same time.The asynchronous sending of the alarm frame interception function is implemented, making the real-time video stream smoother.Auxiliary services are set up to quickly resume video streaming after interruption.

## 5. Future Work

The main contribution of this article is to combine UAV automatic inspection technology and artificial intelligence technology to achieve real-time video stream detection. However, how to design a more complete and efficient UAV video stream detection system still requires more research and also faces huge challenges. For example, more complex backgrounds, how to effectively remove background interference, high-altitude UAVs with smaller viewing angle objects, how to improve the performance of small object detectors, how to make models more lightweight and more suitable for deployment to edge devices, etc.

## Figures and Tables

**Figure 1 sensors-24-03862-f001:**
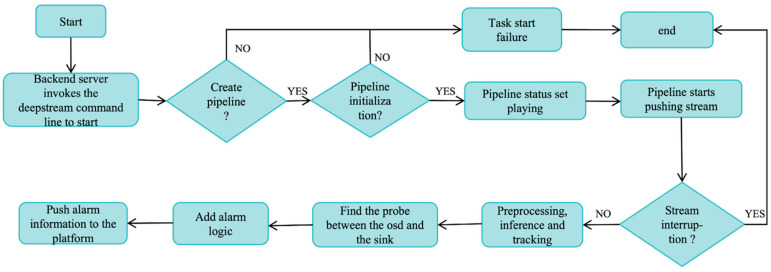
DeepStream architecture diagram. First, the back-end server calls the DeepStream command line to start, and then after a series of flow operations, it finally pushes alarm information to the platform.

**Figure 2 sensors-24-03862-f002:**
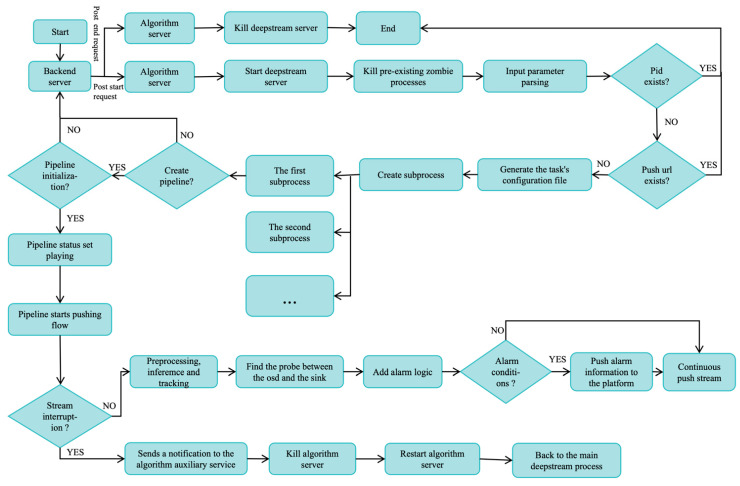
Improved DeepStream architecture diagram. First, start the backend service, then go through more judgment initialization and other operations, and finally go through the backend service feedback.

**Figure 3 sensors-24-03862-f003:**
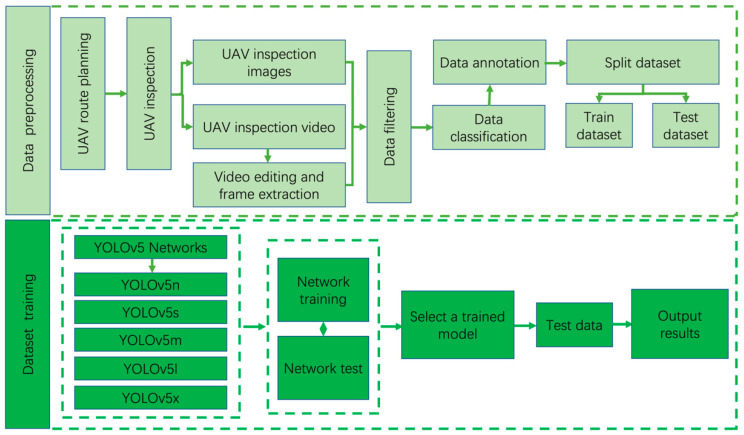
Algorithm training architecture diagram. It consists of two stages: data preprocessing and data training.

**Figure 4 sensors-24-03862-f004:**
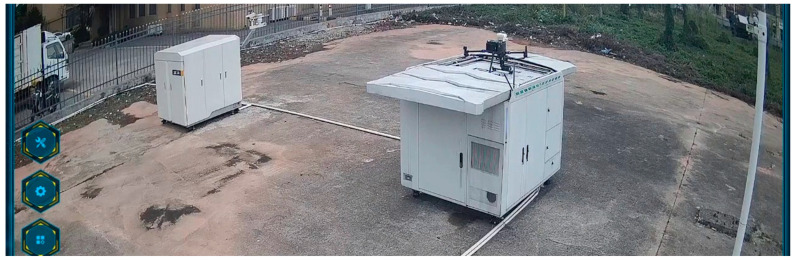
UAV command and dispatch platform. This is a UAV ready to take off displayed on the UAV monitoring platform.

**Figure 5 sensors-24-03862-f005:**
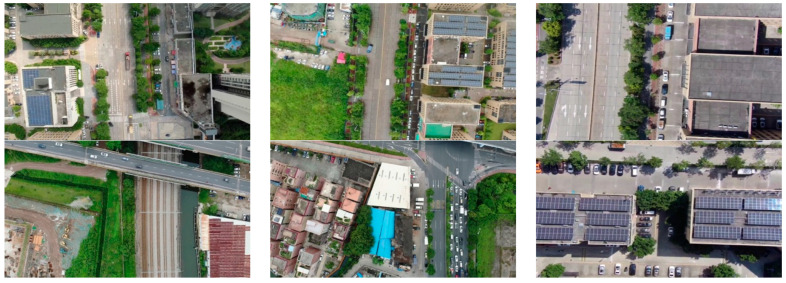
Original image of the data set. Images of different scenes collected by UAV from a high-altitude perspective.

**Figure 6 sensors-24-03862-f006:**
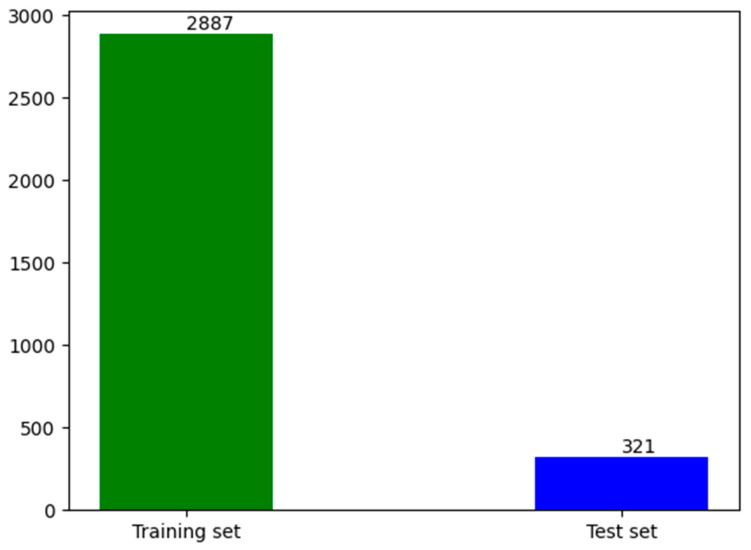
Dataset statistics. In the process of training, the data set, the data set is divided into a training set and a test set.

**Figure 7 sensors-24-03862-f007:**
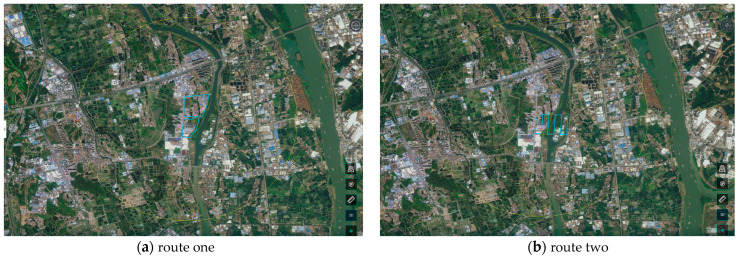
Different route planning. The flight routes of UAV include different areas such as towns and water bodies. The blue line is the planned UAV route.

**Figure 8 sensors-24-03862-f008:**
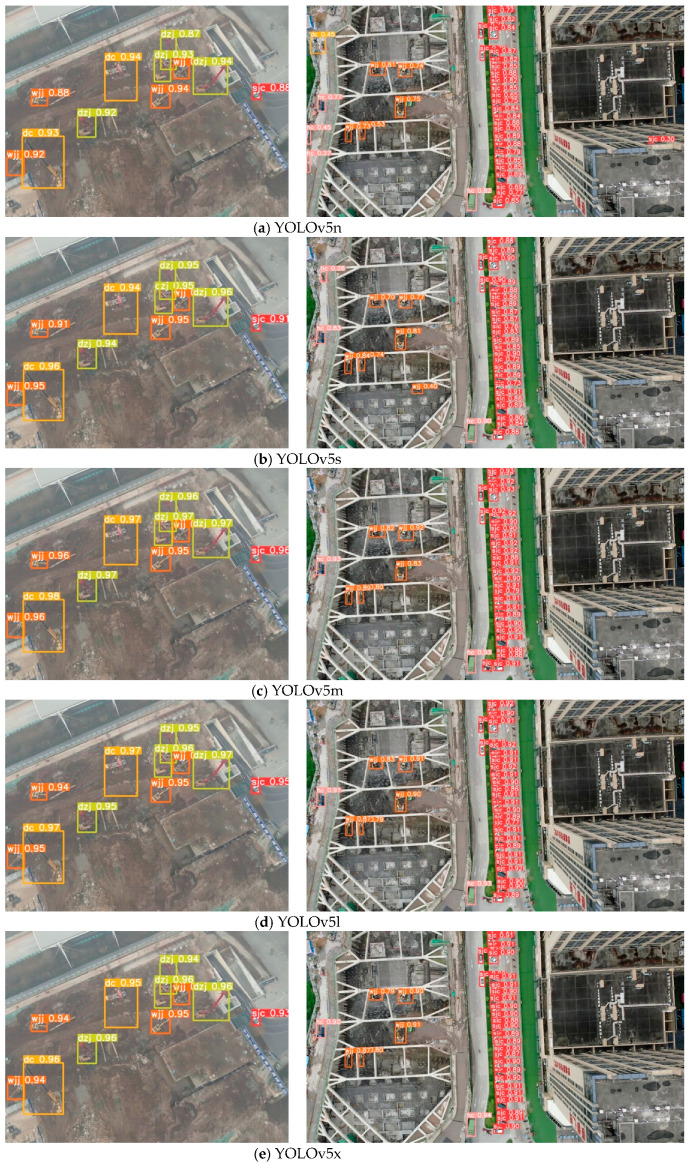
YOLOv5 detection results. The prediction of (**a**) YOLOv5n; (**b**) YOLOv5s; (**c**) YOLOv5m; (**d**) YOLOv5l; and (**e**) YOLOv5x. It shows the detection results of the YOLOv5 model on this data set.

**Figure 9 sensors-24-03862-f009:**
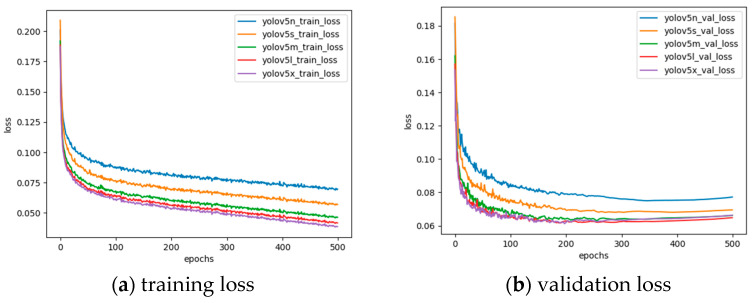
YOLOv5 model loss curve.

**Figure 10 sensors-24-03862-f010:**
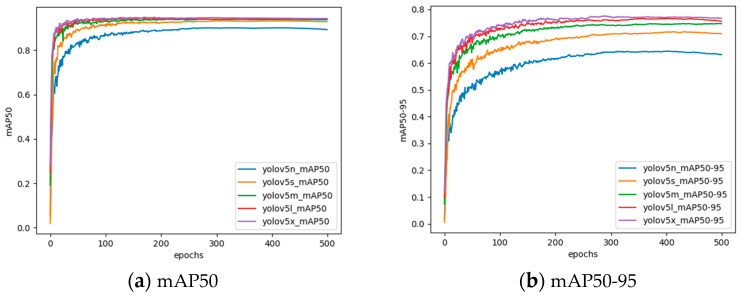
YOLOv5 model accuracy curve.

**Figure 11 sensors-24-03862-f011:**
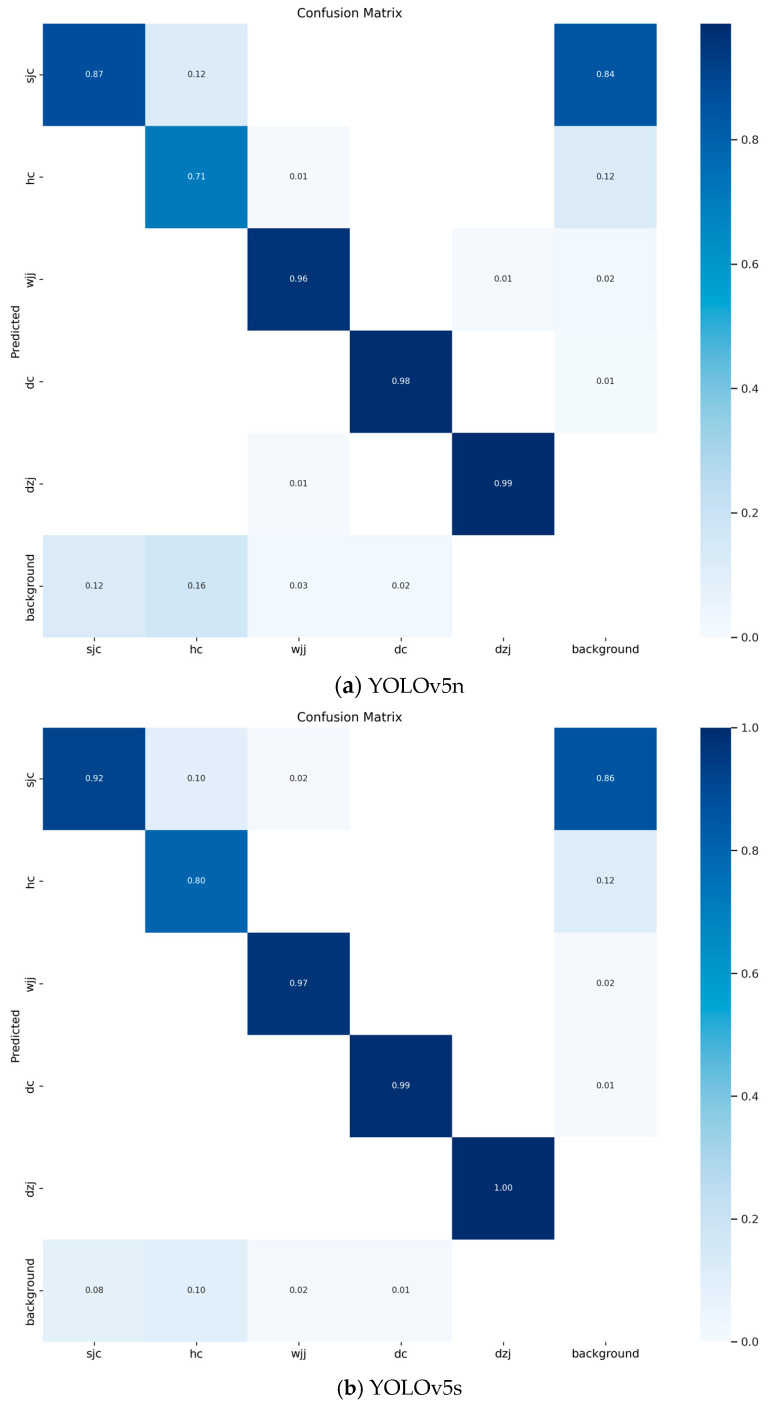
Confusion matrixes.

**Figure 12 sensors-24-03862-f012:**
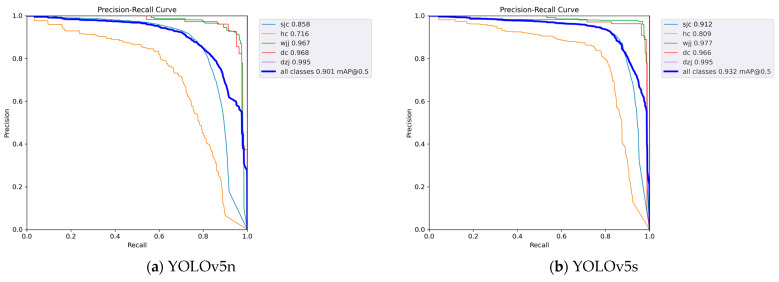
The precision and recall curves.

**Figure 13 sensors-24-03862-f013:**
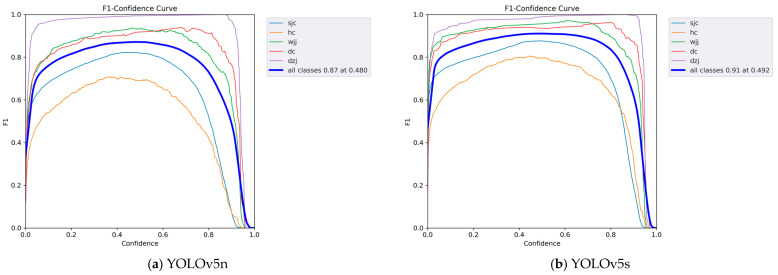
F_1_ value and confidence curve.

**Figure 14 sensors-24-03862-f014:**
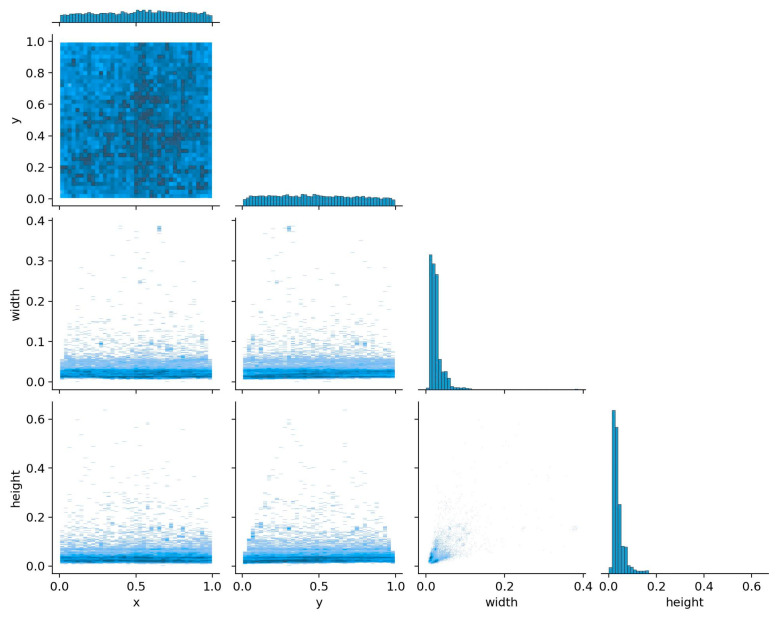
The correlation of labels.

**Figure 15 sensors-24-03862-f015:**
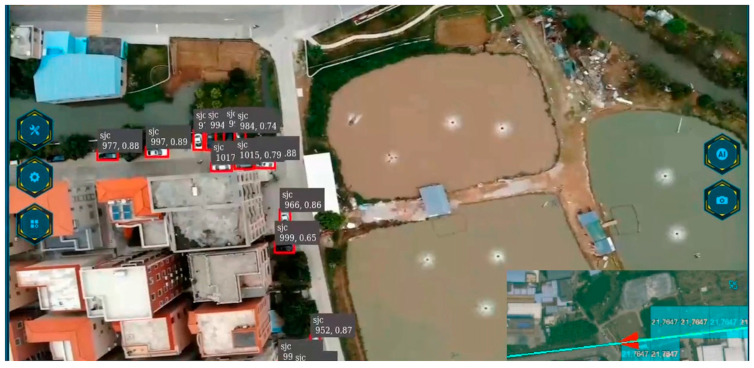
UAV inspection interface. It shows a scene diagram of real-time detection and tracking of high-altitude inspections, and the UAV flight route can be seen in the lower right corner.

**Table 1 sensors-24-03862-t001:** Flight parameters of different routes.

Routes	Flight Length/m	Flight Minutes	Route Altitude/m	Taking off and Landing Height/m	Round Trip Speed/m/s	Route Speed /m/s
One	4150.8	11.07	127	129	10	8
Two	4314.8	17.95	127	131	10	5
Three	4912.8	18.93	150	155	10	5
Four	4728.6	19.42	127	131	10	5

**Table 2 sensors-24-03862-t002:** Comparison of mAP50 detection results of different models.

	sjc/%	hc/%	wjj/%	dc/%	dzj/%	mAP50/%
Yolov5n	85.8	71.6	96.7	96.8	99.5	90.1
Yolov5s	91.2	80.9	97.7	96.6	99.5	93.2
Yolov5m	91.8	83.4	96.7	97.3	99.5	93.7
Yolov5l	93.1	83.7	97.2	97.6	99.5	94.2
Yolov5x	93.2	86.1	96.9	97.8	99.5	94.7

**Table 3 sensors-24-03862-t003:** Comparison of mAP50-95 detection results of different models.

	sjc/%	hc/%	wjj/%	dc/%	dzj/%	mAP50–95/%
Yolov5n	51.7	42.0	67.6	75.1	86.0	64.5
Yolov5s	62.2	54.7	74.7	78.5	88.6	71.7
Yolov5m	66.3	61.1	77.4	79.6	89.6	74.8
Yolov5l	68.9	63.1	79.2	81.3	90.3	76.6
Yolov5x	69.7	65.3	79.3	82.4	91.1	77.6

**Table 4 sensors-24-03862-t004:** Comparison of the inference speed of different models.

	Yolov5n	Yolov5s	Yolov5m	Yolov5l	Yolov5x
Model Size/MB	3.7	13.76	40.27	88.57	165.13
Speed/ms	11.1	11.26	15.77	17.44	20.29

**Table 5 sensors-24-03862-t005:** Startup time test.

	The First Stage/s	The Second Stage/s	Total/s	Clarity
1	2.21	0.60	2.81	Ultra HD
2	2.24	0.28	2.52	Ultra HD
3	2.27	1.24	3.51	UD
4	2.27	0.50	2.77	UD
5	2.19	3.52	5.71	HD
6	2.21	3.42	5.63	HD
7	2.21	3.03	5.24	SD
8	2.29	3.24	5.53	SD
9	2.23	3.40	5.63	Smooth
10	2.22	3.51	5.73	Smooth

## Data Availability

Data are contained within the article.

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
