# Peer review of "Real-Time Object Detection from UAV Inspection Videos by Combining YOLOv5s and DeepStream"

_sensors, 2024, doi:10.3390/s24123862_

Round 1

Reviewer 1 Report

Comments and Suggestions for Authors

One of the hot topics in industrial applications of various kinds is real-time object detection.  This area gets considerable attention in computer vision. Within this area, a very challenging problem is real-time inspection observation of objects on the earth's surface from high-altitude UAVs.

The reviewed paper proposes one of the possible solutions to this problem. Namely, a technology of automatic inspection of ground objects based on the use of YOLOv5 type models is developed for UAVs. As an example of implementation of the proposed technology, training of these models is performed using five datasets that contain images of vehicles as inspected objects. It is shown that the detection time of a single object using the described tools is relatively short. In particular, in one of the examples it amounted to 11.26 ms, which is quite acceptable for many computer vision applications. It should be noted that the effectiveness of the proposed approach is demonstrated on a relatively simple example, but it has the potential to solve more complex problems.

The article is in general adequately structured and presented. It may be useful for researchers in the field of computer vision, especially as applied to the specificity of these tasks due to their implementation on board a UAV.

As a remark on the layout of the paper, it should be noted the unsatisfactory quality of Fig. 11. This figure shows five confusion matrices.  For all these matrices it is almost impossible to understand what they demonstrate. All numbers in the figures are very small and unsharp (blurred). Even at high magnification, it is not possible to read the values of numerical parameters in the cells of these matrices, as well as along the abscissa and ordinate axes.

Author Response

Comments 1: As a remark on the layout of the paper, it should be noted the unsatisfactory

quality of Fig. 11. This figure shows five confusion matrices. For all these matrices it is almost

impossible to understand what they demonstrate. All numbers in the figures are very small and

unsharp (blurred). Even at high magnification, it is not possible to read the values of numerical

parameters in the cells of these matrices, as well as along the abscissa and ordinate axes.

Response 1: Thank you for pointing this out. We have modified the quality of the image and

show that the pixel resolution is larger and clearer

Reviewer 2 Report

Comments and Suggestions for Authors

Below, a list of comments from my end:

-``Unmanned aerial vehicles (UAVs) high-altitude real-time inspection has always been a very challenging task.''

-But why is it very challenging? it is also unclear what are the problems been solved in this work.

-``YOLOv5s object detection'' needs significant improvement!

-The description part of the sections 2.1 and 2.2 need more clear and elaborate explanations.

-Apart from Figs 9-14, each of them needs adequate discussion.

-Mechanism part looks very weak.

-Also provide mathematical analysis to support your contribution your work.

-`Motivation and research gap' is unclear from introduction and related work.

Comments on the Quality of English Language

requires proofreading.

Author Response

Comments 1: -``Unmanned aerial vehicles (UAVs) high-altitude real-time

inspection has always been a very challenging task.''

-But why is it very challenging? it is also unclear what are the problems been

solved in this work

Response 1: Thank you for pointing this out. Therefore, We modify it as

follows:

Because high-altitude inspections are susceptible to interference from different weather

conditions, interference from communication signals and a larger field of view result in a

smaller object area to be identified.

Comments 2: -``YOLOv5s object detection'' needs significant improvement!

Response 2: Thank you for pointing this out. Therefore, We modify it as

follows:

Since the UAV flies at high altitude, it will keep shaking, and the field of view when flying

at high altitude is large, and the shooting area is also large, resulting in a smaller target to

be detected. There are difficulties in using object detection algorithms to detect smaller

object. This article mainly uses YOLOv5s, a lightweight object detection model, combined

with DeepStream architecture to achieve real-time video stream detection. To better

implement a complete set of real-time target detection system architecture, we made a

series of improvements to the original DreamStream and combined with the YOLOv5s

model to achieve target detection. YOLOv5s is designed to detect the goals that require

inspection. It is designed to achieve automatic detection capabilities. The main

improvement part is the design of the overall logical architecture of DeepStream, and the

improved experimental results are also very effective.

Commennt 3: -The description part of the sections 2.1 and 2.2 need more clear and elaborate

explanationsResponse 3: Thank you for pointing this out. Therefore, We modify it as

follows:

As shown in Figure 1, This figure shows the basic workflow of DeepStream’s original architecture.

First, we need to start a DeepStream video streaming service on the backend of the platform,

immediately start creating the DeepStream pipeline, and determine the pipeline initialization status. If

the pipeline initialization verification is successful, then the pipeline status is set to the playing state,

which is the pipeline starting working status. Secondly, the pipeline starts to push the stream, and the

algorithm performs video stream data pre-processing including normalization and scaling. Then the

video stream will be input into the YOLOv5 model for inference, and finally the Deepsort [37,38]

tracking algorithm is used to track the corresponding object. Add alarm logic to the object that needs to

be identified, push the processed video stream to the platform service, and finally display the

identification results in real time on the platform software.

As shown in Figure 2, this figure shows the improved workflow of DeepStream architecture. First, the

platform starts the DeepStream service command by sending an http request. Currently, once the

algorithm service receives the platform request to start the command, it immediately starts the

DeepStream service, starts parsing the request command parameters, determines whether the

parameters meet the requirements, and generates a configuration file according to the corresponding

task. Secondly, after the configuration is successful, start creating a sub-process to perform the task and

create a DeepStream pipeline. Determine whether the pipeline is initialized. After the initialization is

successful, set the pipeline status to the playing state. The pipeline starts pushing and the algorithm

starts working, including pre-processing, reasoning, tracking and other operations. After the object is

detected, alarm logic is added. Finally, the video stream is pushed to the platform service, and the

platform interface displays the stream inferred by the algorithm. In addition, we also added an auxiliary

service, whose purpose is to monitor whether the DeepStream service is disconnected. If a

disconnection occurs, the auxiliary service will immediately kill the task and restart the task

immediately.

As shown in Figure 3, Figure 3 shows two extremely important parts of data processing and data

training. The first is the data processing stage. In the process of collecting data, we need to plan the

UAV inspection route. In the areas that need to be inspected, let the UAV automatically inspect and

collect photos or videos according to the planned route. During the process of collecting images by

UAV, we flew in different time periods, different weather, and different lighting conditions to achieve

better performance of the model. To better make the model more robust to the complex background of

high-altitude inspections. We plan the flight area to include different areas such as villages, water bodies

and towns. The collected UAV inspection videos need to be manually edited and the required object

information retained and extracted into images. Images were manually filtered and labeled using

LabelImg software. Finally, the data is divided into training set and test set.

The second is the model training stage. In the first stage of data processing, the processed data has been

divided into training set and test set. We are in the data collection phase. Effectively planning routes for

different scenarios and collecting data under different conditions fully ensures the richness of model

training samples and can effectively prevent over-fitting during the training process. We evaluate the

performance of YOLOv5n, YOLOv5s, YOLOv5m, YOLOv5l and YOLOv5x models in the process of

UAV high-altitude inspection. The performance is evaluated on the training set and test set divided on

the same data set. Finally, we will select a model with strong robust performance for real-time detection

based on its real-time detection effect.

Commennt 4: -Apart from Figs 9-14, each of them needs adequate discussion.

Response 4: Thank you for pointing this out. For discussion of the additions to

Figures 1 to 3, please refer to the reply to comment 3.

Figure 4, We add some discussion as follows

Through the command and dispatch system interface, the status of the UAV and its slot can be monitored

in real time. We can see a white drone nest, the drone nest door has been opened, the platform has been

raised, and a black UAV is waiting to fly on it. There are three buttons available on the left side of the

command and dispatch system. The main functions include UAV control, landing control, system

functions, power control and emergency control, etc. It can not only command the takeoff and landing

of the drone, but also control the angle and focus of the gimbal. When the drone is flying, the battery

power can be monitored in real time. When an emergency occurs, such as low battery and automatic

hovering, it can also return to home with one click.

Figure 5, We add some discussion as follows

To ensure the effectiveness of model training in this article, we increase the richness of samples as much

as possible. Therefore, in the process of collecting images, we planned routes in different areas and

different route altitudes. While ensuring flight safety, the drone's flight speed and different flight

mileage are set.

Figure 6, We add some discussion as follows

We divide the training set and the test set at a ratio of 9:1. To ensure the effectiveness of the training

model, the divided data sets are independent of each other.

Figure 7, We add some discussion as follows

The routes are all included in a yellow circular area, which is displayed on a map interface. We can

clearly see that there are residential areas and river areas next to the planned routes. To increase the

richness of the sample, our routes span multiple regions to collect images.

Figure 8, We add some discussion as follows

After the model training was completed, to verify the detection effects of different models, we used

images containing five different categories to test the detection performance of five different models.

We can clearly see that different YOLOv5 detection models can detect five different types of objects:

car, truck, excavator, pile driver and crane.

Figure 15, We add some discussion as follows

It can be seen from Figure 5,It is a display of real-time inspection pictures. During this kind of high

altitude inspection by drones, the drones are looking down at the ground angle. Information such as

buildings, ponds and vehicles on the ground can be clearly seen. And it can also be clearly seen that the

YOLOv5s model trained in this article can automatically detect the required targets during the automatic

drone inspection process, and correctly distinguish the target area and the background area. In the lower

right corner of the image, you can obtain the inspection route of the drone and the specific location

information of the drone flying on this route.

Commennt 5: -Mechanism part looks very weak.

Response 5: Thank you for pointing this out. Therefore, We modify it as

follows:

As shown in Figure 2, this figure shows the improved workflow of DeepStream architecture. First, the

platform starts the DeepStream service command by sending an http request. Currently, once the

algorithm service receives the platform request to start the command, it immediately starts the

DeepStream service, starts parsing the request command parameters, determines whether the

parameters meet the requirements, and generates a configuration file according to the corresponding

task. Secondly, after the configuration is successful, start creating a sub-process to perform the task and

create a DeepStream pipeline. Determine whether the pipeline is initialized. After the initialization is

successful, set the pipeline status to the playing state. The pipeline starts pushing and the algorithm

starts working, including pre-processing, reasoning, tracking and other operations. After the object is

detected, alarm logic is added. Finally, the video stream is pushed to the platform service, and the

platform interface displays the stream inferred by the algorithm. In addition, we also added an auxiliary

service, whose purpose is to monitor whether the DeepStream service is disconnected. If a

disconnection occurs, the auxiliary service will immediately kill the task and restart the task

immediately.

Commennt 6:-Also provide mathematical analysis to support your contribution your work.

Response 6: Thank you for pointing this out. Therefore, We modify it as

follows:

3.5.Resource Utilization Analysis

To verify the quality of a framework, we not only consider its algorithm detection capabilities, but also

consider its reasonable utilization of effective resources. Therefore, it is necessary to compare the real

time video streams before and after the improvement to check the resource overhead of the system

architecture. The main comparison before and after improvement is as follows

1.Before the improvement, it was necessary to bind drone equipment and specify execution tasks. In

terms of resource consumption, a Telsa T4 GPU, 16G graphics card server can only bind up to 6 devices

and tasks to be executed. That is, the binding of tasks limits the reasonable utilization of GPU and

memory resources and cannot be dynamically adjusted according to task requirements. In short, the

performance of the hardware before the improvement was very poor.

2.After improvement, our real-time video stream detection system does not need to bind devices and

tasks: tasks are no longer bound, and resource allocation can be dynamically adjusted according to real

time needs. Not only can multiple devices be controlled at the same time, but different tasks can be

switched freely. Therefore, after the improvement, the flexibility and scalability of the system have been

greatly improved.

Commennt 7: -`Motivation and research gap' is unclear from introduction and related work.

Response 7: Thank you for pointing this out. Therefore, We modify it as

follows:

Existing real-time inspection technology, especially the use of fixed camera equipment for monitoring,

will result in a small field of view, poor mobility of the equipment, the inability to realize real-time

switching between different areas, and the existence of large monitoring blind spots and other problems.

UAV high-altitude inspection can just solve the problems of vision and flexibility. Therefore, this article

proposes a method that combines UAV and artificial intelligence real-time detection to implement an

automatic UAV real-time inspection system

Round 2

Reviewer 2 Report

Comments and Suggestions for Authors

Apart from previous round review comments, few comments been well addressed; although still need to work on many major comments.

requires more effort to address the following comments in order to convince this reviewer.

`Motivation and research gap' is unclear from introduction and related work.'

`Also provide mathematical analysis to support your contribution your work.'

`Apart from Figs 9-14, each of them need adequate discussion.'

Comments on the Quality of English Language

can be improved.

Author Response

Response to Reviewer 1 Comments

Thank you very much for taking the time to review this manuscript. Please find

the detailed responses below and the corresponding revisions/corrections

highlighted/in track changes in the re-submitted files.

Comments 1: -`Motivation and research gap' is unclear from introduction and

related work.'

Response 1: Thank you for pointing this out. Therefore, We modify it as

follows:

Haq et al. [37] deployed the DeepStream framework on the NVIDIA Jetson single-board computer to

run deep learning algorithms, especially the YOLO algorithm. It also verified that the DeepStream

framework can run well in virtual machines, especially using Docker, which can further improve the

performance of the model and the portability during the deployment process. Huu et al. [38] proposed

a method based on the NVIDIA DS-SDK architecture, using multiple surveillance camera detection

methods to implement the application of deep learning-based algorithms for vehicle monitoring.

Ghaziamin et al. [39] deployed the object detection model to Nvidia Jetson devices and designed a

passenger counting system. And after edge deployment through Nvidia DeepStream, it improves

efficiency while saving the use of hardware resources. Smink et al. [40] use edge devices combined

with the detection and tracking system of the NVIDIA DeepStream framework to implement a set of

real-time tag reading applications. Qaraqe et al. [41] designed an end-to-end security intelligent

monitoring system that uses the DeepStream Software Development Kit (SDK) for real-time inference,

which can have a significant impact on public safety and crowd management.

Comments 2: -`Also provide mathematical analysis to support your contribution

your work.'

Response 2: Thank you for pointing this out. Therefore, We modify it as

follows:

DeepStream startup time mainly consists of two parts of time. One is the time from the request to the

pipeline initialization, and the other is the time from the pipeline initialization to the pipeline state

switching to the playing state. The sum of the two is the startup time. The initial application of the

DeepStream service has a startup time of approximately 7 seconds. Therefore, there is huge room for

improvement. Based on this, we test the startup of video streams in different definitions.The

definition is divided into five levels, namely ultra-high definition (Ultra HD), ultra definition (UD), high

definition (HD), standard definition (SD) and smooth. As shown in Table 5.

Table 5. Startup time test.

The first

stage /s

The second

stage /s

Total/s

Clarity

1 2.21 0.60 2.81 Ultra HD
2 2.24 0.28 2.52 Ultra HD
3 2.27 1.24 3.51 UD
4 2.27 0.50 2.77 UD
5 2.19 3.52 5.71 HD
6 2.21 3.42 5.63 HD
7 2.21 3.03 5.24 SD
8 2.29 3.24 5.53 SD
9 2.23 3.40 5.63 Smooth
10 2.22 3.51 5.73 Smooth

As shown in Table 5, during the first test, the first-stage startup time of UHD was 2.21 seconds, and the

second-stage startup time was 0.60 seconds. The total time consumed is 2.81 seconds. The total time

for the second test of UHD video was 2.52 seconds. The overall consumption time is significantly faster

than other low-definition videos. Moreover, the startup time of the optimized model architecture is

significantly faster than the original DeepStream processing time.

Commennt 3: `Apart from Figs 9-14, each of them need adequate discussion.'

Response 3: Thank you for pointing this out. Therefore, We modify it as

follows:

Figure 1. DeepStream architecture diagram. First, the back-end server calls the deepstream command line to start, and

then after a series of flow operations, it finally pushes alarm information to the platform.

Figure 2. Improved DeepStream architecture diagram. First, start the backend service, then go through more judgment

initialization and other operations, and finally go through the backend service feedback.

Figure 3. Algorithm training architecture diagram. It consists of two stages: data preprocessing and data training.

Figure 4. UAV command and dispatch platform. It is a UAV ready to take off displayed on the UAV monitoring

platform.

Figure 5. Original image of the dataset. Images of different scenes collected by UAV from a high-altitude perspective.

Figure 6. Dataset statistics. In the process of training the data set, the data set is divided into a training set and a test

set.

Figure 7. Different route planning. The flight routes of UAV include different areas such as towns and water

bodies.

Figure 8. YOLOv5 detection results. The prediction of (a) YOLOv5n (b) YOLOv5s (c) YOLOv5m (d) YOLOv5l and (e)

YOLOv5x. It shows the detection results of the YOLOv5 model on this dataset.

Figure 15. UAV inspection interface. It shows a scene diagram of real-time detection and tracking of high-

altitude inspections, and the UAV flight route can be seen in the lower right corner.
